# Fabrication of Nitrogen-Doped Carbon@Magnesium Silicate Composite by One-Step Hydrothermal Method and Its High-Efficiency Adsorption of As(V) and Tetracycline

**DOI:** 10.3390/ma16155338

**Published:** 2023-07-29

**Authors:** Xuekai Wang, Jinshu Wang, Jianjun Li, Yucheng Du, Junshu Wu, Heng He

**Affiliations:** 1Key Laboratory of Advanced Functional Materials, Ministry of Education, Faculty of Materials and Manufacturing, Beijing University of Technology, Beijing 100124, China; wangxuekai@aust.edu.cn (X.W.);; 2School of Materials Science and Engineering, Anhui University of Science and Technology, Huainan 232001, China

**Keywords:** diatomite, adsorption capacity, N-doped carbon

## Abstract

Tetracycline (TC) and arsenic contaminants are two main pollutants in aquaculture and livestock husbandry, and they have drawn worldwide attention. To address this issue, a novel N-doped carbon@magnesium silicate (CMS) was fabricated via a facile and low-cost hydrothermal route, adopting glucose and ammonia as C and N sources, respectively. The synergetic combination of carbon and magnesium silicate makes CMS possess a high surface area of 201 m^2^/g and abundant functional groups. Due to the abundant C- and N-containing functional groups and Mg-containing adsorptive sites, the maximum adsorption capacity values of CMS towards As(V) and TC are 498.75 mg/g and 1228.5 mg/g, respectively. The type of adsorption of As(V) and TC onto CMS is monolayer adsorption. An adsorption kinetic study revealed that the mass transfer and intraparticle process dominates the sorption rate of As(V) and TC adsorption onto CMS, respectively. Various functional groups synthetically participate in the adsorption process through complexion, π–π EDA interactions, and hydrogen bonds. This work provides a one-step, low-cost route to fabricate a N-doped carbonaceous adsorbent with a high surface area and abundant functional groups, which has great potential in the application of practical sewage treatment.

## 1. Introduction

Nowadays, due to the increasing demand for meat products, aquaculture and livestock husbandry have developed rapidly. Serious environmental issues have been caused by unreasonable development in the above areas, which seriously affects the biological environments of aquatic ecosystems [1,2]. Tetracycline (TC) and arsenic species are two of the main pollutants detected in sewage discharged from livestock ranches and intensive fishing grounds [3,4]. TC is an efficient antibiotic, which is widely used for the prevention and cure of diseases in fish and livestock [5,6]. However, TC is quite difficult to metabolize. Long-term exposure to TCs can lead to an increase in bacterial resistance, thus posing a great threat to human health and ecological environments [7,8]. Similarly, arsenic exists as heavy metal oxyanions and is commonly used as a growth promoter in animals and as a pesticide. The enrichment of arsenic species through migration can cause serious damage to human health and ecosystems [9]. Thus, great attention has been paid to the treatment of sewage containing these aforementioned pollutants. Various strategies have been adopted for the removal of TC- and arsenic-containing sewage, including biological treatment, chemical precipitation, photocatalysis, Fenton catalysis, adsorption, etc. Among these methods, adsorption is commonly recognized as the most efficient and facile method due to its wide applicable scope, facile operation, low cost, and convenience for field application.

Various adsorbents have been developed. Recently, carbonaceous adsorbents such as graphene oxide, activated carbon, and biochar have attracted wide attention due to their large specific surface area, abundant C-containing functional groups, and good adsorption performance [10,11,12,13]. Furthermore, with the doping of specific elements such as N and S, the adsorption performance of carbonaceous adsorbents can be effectively improved, since the doping of N and S elements can increase the amount of specific surface-adsorptive sites [14,15,16,17,18]. In recent years, carbonaceous adsorbents have shown great application potential in the removal of TC- and arsenic-containing sewage.

Although carbonaceous adsorbents exhibit good adsorption performance in sewage treatment, the reported carbonaceous adsorbents are far from meeting the requirements of practical environment remediation. The complicated separation process of highly dispersed carbonaceous adsorbents in water could cause secondary pollution to the environment. Furthermore, raw carbonaceous adsorbent commonly needs an extra activation process to obtain a higher specific surface area, and a carbonaceous composite is commonly obtained by a two-step fabrication process [19,20]. Moreover, a further calcination treatment under a nitrogen-containing atmosphere is also required for the N-doping of carbonaceous adsorbent [16,18]. This leads to a complicated fabrication process with intrinsic environmental hazards and energy consumption. Therefore, the facile fabrication of a carbonaceous adsorbent with a high specific surface area still remains a challenge.

To address this issue, a diatomite template self-sacrificing method is proposed. As reported in our previous research, diatomite can be adopted as a self-sacrificing template in the synthesis of 3D hierarchical porous magnesium silicate with a high specific surface area of 337 m^2^/g [21]. The obtained magnesium silicate inherits its structural characteristics from diatomite and offers substantial interfaces for the guest product. Herein, with diatomite as a self-sacrificing template, magnesium silicate and hydrothermal carbon could be synthesized through one hydrothermal route. The combination of hydrothermal carbon and magnesium silicate provides the composite with a high surface area and abundant C-containing groups. Ammonia used in the hydrothermal route offers hydroxyl bonds for the formation of magnesium silicate and also serves as a N source for the formation of N-doped carbon. Notably, the Mg-adsorptive site can promote the removal of TC through a cation bonding bridge, i.e., a cation–n bond and a cation–p bond [22]. Thus, the design of carbon@magnesium (CMS) offered an ideal solution to obtain a N-doped carbonaceous composite with a high specific surface area and abundant C and N groups with a facile one-pot hydrothermal route. Owing to the high specific surface area and abundant functional groups, the obtained N-doped CMS exhibited excellent adsorption capacities for TC (1228.5 mg/g) and arsenic (498.75 mg/g). It was found that Mg-containing adsorptive sites and C- and N-containing groups synergistically participated in the removal of As(V) and TC. The adsorption kinetics revealed the difference in the adsorption process between As(V) and TC by CMS. Furthermore, the adsorption mechanisms of CMS towards TC and As(V) were also discussed. Above all, this work provides a strategy for the facile one-pot synthesis of N-doped carbonaceous magnesium silicate for the efficient removal of TC and As(V) from sewage.

## 2. Experimental

### 2.1. Materials and Chemicals

Glucose (C_6_H_12_O_6_), magnesium chloride hexahydrate (MgCl_2_·6H_2_O), hydrochloric acid (HCl, 1 mol/L), sodium hydroxide solution (NaOH, 8 g/L), ethanol (C_2_H_5_OH), ammonia (25% wt%), and TC (C_22_H_24_N_2_O_8_) were purchased from Beijing Chemical Reagents Company. All the chemicals were analytical regent grade and used without further purification. An arsenic solution of l g/L was purchased from the National Research Center for Certified Reference Materials. Diatomite was obtained from Changbai, Jilin province.

### 2.2. Sample Synthesis

The preparation of CMS was accomplished based on our previous work with some modification [21]. Typically, 1 g glucose and 1g raw diatomite were added into a mixed solution of 10 mL deionized water and 40 mL ammonia solution (25 wt%). Then, the suspension was kept stirring for 15 min, and 5 mL ethanol was added to the suspension while stirring. Next, 5 g MgCl_2_·6H_2_O was added into 20 mL deionized water. The obtained MgCl_2_ aqueous solution was dropwise added to the obtained suspension under stirring for 30 min. The final suspension was transferred to a reaction caldron with hydrothermal treatment at 180 °C for 8 h. The precipitates were rinsed 3 times with deionized water and ethanol and dried at 60 °C for 3 h. The detailed synthesis processes of raw materials and product are listed in Table 1.

### 2.3. Materials Characterization

The morphology and element distribution of samples were observed using a field emission scanning electron microscopy (Gemini SEM 300, ZEISS, Oberkochen, Germany) and transmission electron microscopy (FEI Talos F200X-G2, FEI, Boston, MA, USA). Phase composition was measured on a D8 Advance X-ray diffractometer using Cu Kα1 radiation. The specific surface area and pore size distribution were measured using an ASAP 2020 (Micromeritics, Atlanta, GA, USA) apparatus utilizing BET and BJH methods. The functional groups and bonds were identified with a Perkin-Elmer 1730 spectrometer (PerkinElmer, Waltham, MA, USA) using the KBr pressed disk method. X-ray photoelectron spectroscopic (XPS) spectra were measured with an ESCALAB 250Xi electron spectrometer (Thermo Fisher, Waltham, MA, USA). The UV–vis spectrum of the TC solution was recorded using a UV-3600 spectrophotometer (SHIMADZU, Kyoto, Japan). The concentration of arsenic solution was measured using the ICP-AES technique with ICP-AES_OPTIMA7000DV (PerkinElmer, Waltham, MA, USA) equipment with a detection limit of 0.001 mg/L. The pH of solutions was measured with a PHS-3C pH meter (INESA, Shanghai, China). ^13^C solid-state magic angle spinning (MAS) NMR experiments were conducted using a Bruker Advance 400 MHz WB spectrometer (Bruker, Karlsruhe, Germany) using 4 mm zirconia rotors as sample holders spinning at a MAS rate of ν_MAS_ = 15 kHz. Thermo-gravimetric measurements were conducted using a differential scanning calorimeter (DSC 204F1, NETZSCH, Selb, Bavaria, Germany).

### 2.4. Adsorption Capacity and Removal Efficiency Measurement

Both experiments were conducted to investigate the adsorption isotherms and kinetics. For adsorption isotherms, 0.02 g of the prepared CMS was added to 70 mL As(V) and TC solutions with different concentrations (50–800 mg/L), which were then treated under stirring for a certain time at room temperature. As for the adsorption kinetics study, 0.02 g CMS was added to 50 mL As(V) and a TC solution with a concentration of 100 mg/L. The mixture was kept under stirring at 500 rpm. At predetermined time intervals (1–5 min), the mixture was withdrawn and filtered for measurement. A syringe filter with a hydrophilic PTFE membrane (diameter = 0.22 μm) was used to filter the mixed solution. HCl or NaOH aqueous solutions were used to adjust the pH. The concentrations of As(V) species and TC in the filtrates were measured by ICP–AES and UV–vis techniques, respectively.

The As(V) and TC equilibrium adsorption capacity (*Q_e_*) and removal efficiency (*R_e_*) were calculated according to the following formulas:(1)$$Q_{e}=\frac{(C_{0}-C_{e})V}{m}$$
(2)$$R_{e}=\frac{C_{0}-C_{e}}{C_{0}}\times 100\%$$
where *C*_0_ and *C_e_* represent the initial and equilibrium concentrations of As(V) and TC species in solution, respectively. *V* represents the volume of the solution, and *m* is the mass of the adsorbent.

## 3. Results and Discussion

### 3.1. Phase Composition of the Samples

Figure 1 shows the XRD patterns of the obtained carbonaceous materials with different raw material ratios. As summarized in Table 1, no significant diffraction peaks of magnesium silicate were observed, as seen in Figure 1a, with only 20 mL ammonia added in the fabrication aqueous solution. The diffraction peaks at 2*θ* = 18.6°, 38.0°, 50.8°, 58.6°, 62.0°, 68.2°, and 72.0° corresponded to (001), (011), (012), (110), (111), (103), and (201) crystal planes of Mg(OH)_2_ with the Brucite phase (JCPDS# 83-0114), respectively, whereas with abundant ammonia (40 mL) added, the diffraction peaks at 2*θ* = 6.1°, 12.3°, 18.5°, 19.4°, 19.4°, 34.2°, 35.8°, and 60.0° corresponded to (002), (004), (006), $\left(1\bar{1}1\right)$, $\left(\bar{1}\bar{1}1\right)$, (132), $\left(1\bar{3}4\right),$ and (060) crystal planes of Mg_6_Si_4_O_10_(OH)_8_ with the chlorite phase (JCPDS# 73-2376), respectively, as illustrated in Figure 1b–d [22]. Additionally, no observable differences in XRD patterns can be detected in Figure 1b–d. The above results revealed that the donated amount of ammonia, rather than glucose or ethanol, in the fabrication process had an obvious influence on the phase composition of the product fabricated.

Although the XRD technique was sufficient to characterize the phase of the magnesium silicate component, no peaks of carbonaceous adsorbent were detected. In order to confirm the presence of carboxylic groups in CMS, the ^13^C-NMR technique was employed to characterize the C species, and the spectrum is illustrated in Figure 2. Two large peaks existed at chemical shifts of 16–60 ppm and 100–150 ppm, corresponding to aliphatic ether carbons and sp^2^-hybridized carbons, respectively. The latter peak revealed a large number of C=C bonds in fabricated carbonaceous adsorbent [23], while the peak at around 175 ppm referred to carboxylic acids (COOH) and ester groups (COOR) [24]. Theses peaks indicated abundant C- and O-containing groups in the fabricated carbonaceous adsorbent, which was favorable for the removal of contaminants by carbonaceous CMS. Still, small differences existed between CMS and pure C (Table 1, C1). The intensity of peaks in CMS (Figure 2E) was lower than in pure C, CMH, CMS-1, and CMS-2. According to the fabrication process (Table 1), the donated amount of glucose in CMS was much lower than in other samples, resulting in a low content of carbonaceous species in CMS. This was the main reason for the low peak intensity in Figure 2E.

According to the results of XRD and ^13^C-solid NMR, the main components in the composite CMH were magnesium hydroxide and hydrothermal carbon, whereas CMS, CMS-1, and CMS-2 were mainly composed of magnesium silicate and hydrothermal carbon. The carbonaceous species provided functional C-containing groups, and thus the TG-DSC technique was employed to measure the content of carbonaceous species in the composite. As illustrated in Figure 3, the mass loss process can be divided into four stages. During the first stage, the water molecules associated with interparticle and interlayer surfaces are removed completely before 200 °C [25]. During the third stage, structural water is removed with a significant endothermic peak at 350 °C, while the mass loss in the second and last stage is ascribed to the decomposition of carbonaceous species. Thus, the content of carbonaceous species in the composite is the mass loss in the second and last stage [25].

Regarding the above analysis, the contents of carbonaceous species in samples CMH, CMS-1, CMS-2, and CMS were 24.2%, 15.0%, 21.4%, and 11.4%, respectively. The C content of CMH and CMS-2 was approximately two times that of CMS, suggesting that the glucose-donating amount had a positive effect on the C content of the fabricated carbonaceous adsorbent.

### 3.2. Surface Morphologies and Pore Structures

To obtain a comparative morphology of CMS, the morphology of pure carbon was firstly characterized and is illustrated in Appendix A. Without ethanol added in the fabrication process, the morphology of pure carbon was composed of membranes and microspheres with a diameter of 8 μm (Appendix A), while the obtained carbonaceous materials existed only as microspheres with a diameter of 4 μm (Appendix A) when ethanol was used in the fabrication process. Moreover, the microspheres were uniformly distributed without agglomeration.

Figure 4 shows SEM images of the samples obtained under different fabrication processes. As observed in Figure 4A,B, nanospheres with diameters of 200–250 nm were scattered on the edge of nanoflowers. Increasing the amount of ammonia to 40 mL, one could see that the obtained structure of CMS-1 had a disk-like morphology with uniform nanospheres on its surface (Figure 4C,D). With ethanol being added into the fabrication solution, the obtained morphology exhibited a significant change. As presented in Figure 4E,F, numerous urchin-like structures were observed on the surface of the disk-like structure. Further decreasing the amount of glucose added, the obtained CMS presented a unique 3D structure (Figure 4G,H), similar to that of diatomite. Numerous flower spheres were dispersed on the outer surface of CMS, and these flower spheres were assembled by nanopetals (Figure 4H), whereas the inner structure of CMS was similar to that of the Mg-chlorite reported in our previous work [21]. The inner structure of CMS consisted of nanocolumn arrays and cavity structures built by columns on the upper and lower surfaces (Appendix A). These nanocolumn arrays grew in the pores of the self-sacrificing template diatomite and prevented the 3D structure from collapsing.

According to XRD and NMR analyses, CMS was composed of a carbonaceous phase and magnesium silicate. It was essential to determine the relationship between morphology and substance. Thus, the TEM technique was adopted to exhibit a visual expression of the above relationship. As presented in Figure 5A, the C and N elements shared the same element distribution, which implies that the spheres were mainly carbon spheres. However, the Mg and Si elements distributed in accordance with the shape of the nanopetals, indicating that the flower sphere in CMH was mainly Mg(OH)_2_ according to the XRD analysis. Moreover, the Si element distribution was not as clear as that of the Mg element, indicating that little magnesium silicate existed in CMH. The element distribution of CMS-1 is presented in Figure 5B, and the C element distribution revealed that the nanospheres in CMS-1 were also carbon spheres. Figure 5C,D show TEM images and element distributions of samples CMS-2 and CMS, respectively. C, N, Mg, and Si elements scattered similarly, indicating that the morphology of CMS-2 and CMS was constructed by both carbon and magnesium silicate. Taking the fabrication process into account, the donation of ethanol improved the distribution of the above four elements.

### 3.3. Morphology Formation Mechanism of Carbon Species

It is generally recognized that the growth mechanism of hydrothermal carbon is similar to that of inorganic crystals. The growth of pure hydrothermal carbon comprises homogeneous nucleation and crystal growth. In a typical hydrothermal fabrication process, glucose is pyrolyzed into organic acids such as formic acid, acetic acid, and acrylic acid. With the pyrolyzation reaction proceeding, glucose is finally converted into the intermediate hydroxymethyl furfural (HMF). Through a series of complicated reactions such as polymerization and condensation, nucleation is finally accomplished [26]. Then, the carbonaceous growth unit comes to the nucleus by diffusion and adsorption. Carbonaceous nuclei grow uniformly and isotropically into carbon spheres, as described by the LaMer nucleation diffusion control model [27].

In the fabrication process of CMS, the generated Mg(OH)_2_ and diatomite can serve as a substrate for the nucleation of hydrothermal carbon at the initial stage. Thus, the nucleation of hydrothermal carbon is a heterogeneous nucleation process. Due to the lower energy required for heterogeneous nucleation, numerous carbonaceous nuclei are generated on the surface of the substrates during the initial stage, while in the final stage, the continuous isotropic growth of nanospheres is limited by the decreasing concentration of the carbonaceous growth unit, leading to a nanosphere morphology of carbon species in CMS. Once the organic solvent ethanol is donated to the hydrothermal solution, the polarity and surface tension decrease, which helps to improve the solubility of the generated oligomer and thus prevents the formation of an ordinary morphology [27,28]. The oligomer is adsorbed on the surface of magnesium silicate and exists as amorphous carbon [29].

### 3.4. Surface Areas and Pore-Size Distributions of the Samples

The specific surface area is a key element of an inorganic adsorbent in order to accomplish good adsorption performance. Figure 6 exhibits the N_2_ adsorption–desorption isotherms and pore-size distributions of the fabricated carbonaceous materials. The four samples shared a similar type IV N_2_ adsorption–desorption isotherm due to their mesoporous structures. Moreover, the H3 hysteresis loop could be observed in the relative pressure range of 0.4–1.0, which was ascribed to slit-shaped and wedge-shaped pore structures due to the non-rigid aggregation of flakes [21]. Additionally, slit-shaped and wedge-shaped pore structures as well as nanopetals could be seen, as shown in Figure 4. Thus, it could be concluded that the nanopetal structures in the fabricated samples contributed most to the specific surface area. The insets in Figure 6 show the detailed pore-size distribution curve of each sample. The pore sizes were in the range of 0–120 nm and centered at 0–20 nm, which also confirmed the existence of mesoporous structures in the fabricated CMS.

As summarized in Table 1, the specific surface areas of the fabricated CMH, CMS-1, CMS-2, and CMS were 32 m^2^/g, 27.7 m^2^/g, 33.7 m^2^/g, and 201 m^2^/g, respectively. Through the comparison of these specific surface areas, it could be concluded that the donated amount of glucose had a significant effect on the surface area of the fabricated samples. As reported in our previous study, 3D magnesium silicate fabricated in a similar route possessed a high specific surface area of 337 m^2^/g, which was obviously higher than that of the fabricated carbonaceous CMS [21]. The large decrease in specific surface area suggested that the carbonaceous component in the CMS compound had a negative effect on the surface area. Additionally, with a decrease in the amount of glucose addition, CMS exhibited a specific surface area of 201 m^2^/g, which was significantly higher than that of CMH, CMS, and CMS-1.

Nanospheres adsorbed on the petal structure in CMH and CMS-1, blocking their pore structures and decreasing their surface areas. However, when ethanol was added in the fabricated process, the hydrothermal carbon was in the amorphous form and was adsorbed on the surfaces of CMS-2 and CMS. The amorphous carbon covered the surface of magnesium silicate and blocked the pore structures, resulting in a low surface area of CMS-2. With a decrease in glucose dosage, the surface area of CMS exhibited an observable increase compared to that of CMS-2. Three reasons mainly contributed to the higher surface area of CMS. Firstly, CMS possessed a unique 3D structure constructed by nanospheres, nanocolumn arrays, and a cavity structure, which was favorable to a high surface area. Secondly, the nanosphere and column structure were assembled by nanopetals, which led to a numerous piled pore structure. Lastly, with the decrease in glucose donation, the blocked pore structures were decreased, resulting in a more exposed pore structure. Thus, CMS exhibited a higher specific surface area than CMH, CMS-1, and CMS-2.

### 3.5. Adsorption Capacity of CMS towards As(V) and TC

Surface area is a key element of the adsorption performance of the fabricated inorganic adsorbent. CMS exhibited a much higher specific surface area than other fabricated carbonaceous adsorbents in this work. Thus, CMS was selected as an adsorbent for the removal of TC and As(V). Before conducting the adsorption experiments, it was essential to determine the zeta potential of the adsorbent and the distributions of contaminants at different pH values. As illustrated in Appendix A, the pH_zpc_ of CMS was about 3, indicating that CMS was negatively charged in solution at pH = 5–10. As for contaminants, TC mainly existed as TCH^−^ and TCH_2_ at pH = 5–10 [22], while As(V) existed as H_2_AsO_4_^−^ and HAsO_4_^2−^ at pH = 5–10 [22]. To simulate a live environment of adsorption conditions in sewage treatment, the pH of aqueous solutions was set at 7.

Figure 7 shows equilibrium adsorption capacities of different samples towards As(V) (A) and TC (B). As presented in Figure 7A, the equilibrium adsorption capacity of CMS increased with the initial concentration of As(V) until it reaches an equilibrium. In the initial concentration of 600 mg/L, the experimental maximum adsorption capacity of CMS towards As(V) was 498.75 mg/g. Notably, the maximum adsorption capacity of CMS towards As(V) was higher than that of magnesium silicate (201 mg/g) and pure carbon (12 mg/g) fabricated in a similar manner, suggesting that the combination of magnesium silicate with hydrothermal carbon possesses better adsorption properties than a single phase. Similar phenomena were observed in the adsorption of TC (Figure 7B). The maximum adsorption capacity of CMS towards TC was 1228.5 mg/g, which was higher than that of magnesium silicate (314 mg/g) and pure carbon (21.5 mg/g). This provided additional evidence of the advantage of a combination.

For the adsorption of As(V) and TC onto CMS, two factors limited its consistent increasing in adsorption capacity with an increase in its initial concentration. Firstly, with limited surface area, surface-adsorptive sites became insufficient, as the initial concentrations of the contaminant kept increasing, and the adsorption process gradually came to an equilibrium state. Secondly, the adsorption of contaminants onto CMS decreased its surface energy and made CMS less adsorptive to contaminants. Finally, the adsorption capacity of CMS reached a saturation value. Notably, in a relatively low concentration of TC, one TC molecule tended to connect with two Mg-adsorptive sites on the surface of CMS [30], whereas in a relatively high concentration of TC, one TC molecule connected with one Mg-adsorptive site [30]. This phenomenon was ascribed to the adsorption nature of CMS towards TC, which also confirmed the strong connection between Mg-adsorptive sites and TC molecules.

Furthermore, recyclability is an important factor for practical sewage treatment. To demonstrate its reusability, the adsorbent was washed with HCl solution (1 mol/L) and deionized water after each adsorption procedure several times. The experimental results of CMS reusability are presented in Appendix A. After the fourth adsorption cycle, the adsorption capacity of CMS towards As(V) and TC decreased from 243.3 to 163.1 mg/g and from 395.5 to 298.3 mg/g, respectively. Appendix A displays the XPS survey of CMS after the first and fourth adsorption cycles. As illustrated in Appendix A, after the first and fourth adsorption cycles, the XPS signals of C 1s, N1s, and Mg 1s experienced a significant drop in peak intensity compared to signals of CMS. The decreases in C, N, and Mg signals partly suggested a decrease in adsorptive sites on the surface of the regenerated adsorbent CMS compared with raw adsorbent CMS. A similar phenomenon was observed in Appendix A. However, the peak intensity of C 1s exhibited an obvious increase. As summarized in Appendix A, the atomic ratios of the C element in CMS and CMS after the first and fourth TC adsorption cycles were 11.25%, 35.45%, and 61.32%, respectively. The increase in the atomic ratio was consistent with the change in the C 1s peak intensity. Two factors mainly contributed to the increase in the C 1s peak intensity. Firstly, as TC mainly consists of carbonaceous material, the adsorption of TC onto CMS surely increased the C1s peak intensity. Secondly, after four TC adsorption cycles, the TC contaminant adsorbed onto CMS remained on the surface of CMS after regeneration treatment. The decrease in the adsorption capacity of CMS was due to the insufficient recovery of adsorptive sites and partial irreversible adsorption of CMS towards As(V) and TC. Overall, it can be concluded that the adsorbent CMS can be used as a potential adsorbent in practical applications due to its high adsorption capacity and good reusability.

To evaluate the adsorption properties of CMS, the adsorption capacities of CMS towards As(V) and TC were compared with the as-reported results, as summarized in Table 2. As illustrated, the fabricated CMS exhibited a higher maximum adsorption capacity towards TC than the reported common adsorbents, i.e., magnetic graphene oxide [31], N-doped carbon [32], Y-GO-SA [22], Fe-doped activated carbon [33], and boron nitride with N-defects [34] but a slightly lower capacity than Fe/porous carbon [35]. As for the removal of As(V), the maximum adsorption capacity of CMS was lower than that of Mg–N-co-doped lignin [36], and covalent organic frameworks [37] but higher than Y-GO-SA [22] and Yttrium-doped iron oxide [38]. Therefore, CMS exhibits a higher adsorption capacity towards TC and a good adsorption capacity towards As(V). Regarding the facile and low-cost fabrication route, CMS possesses great advantages over common adsorbents in practical environmental remediation.

### 3.6. Adsorption Isotherms and Kinetics

For an adequate reflection of the affinity between CMS and As(V)/TC, adsorption isotherms of Freundlich, Langmuir, and Temkin models were chosen to describe their adsorption behaviors. The linear equations of Langmuir, Freundlich, and Temkin models are summarized in Appendix A. The adsorption isotherms of As(V) and TC on CMS studied at 298 K are shown in Appendix A. The experimental adsorption data of As(V) and TC fit well with the Langmuir model, with higher correlation coefficients of 0.996 and 0.999, respectively. The fitting results imply that the adsorption of As(V) and TC are both monolayer adsorption [30]. Adsorptive sites and functional groups on the surface of CMS rather than electrostatic attraction play major roles in the removal of As(V) and TC. The Temkin isotherm fitting results suggest that the adsorption of As(V) and TC onto CMS is an exothermic process [21].

As for the kinetics study, the effect of adsorption time on the equilibrium adsorption capacity of CMS towards As(V) (A) and TC (B) is plotted in Figure 8. The adsorption of As(V) onto CMS was accomplished within 30 min, whereas, the equilibrium adsorption of TC onto CMS required 180 min. Differences in equilibrium adsorption times revealed that diverse adsorption mechanisms occurred during the adsorption of As(V) and TC by CMS.

For an adequate appreciation of the As(V) and TC adsorption processes onto CMS, kinetic models including pseudo-first-order (PFO) and pseudo-second-order (PSO) were employed to interpret the adsorption plots originated from kinetic adsorption experiments (detailed information about the kinetics models is listed in Appendix A). The linearly fitting curves of PFO and PSO are illustrated in Appendix A, and the results are summarized in Table 3. The coefficient (R^2^) of the PSO model for both As(V) and TC adsorption was nearly 1, which was higher than that of the PFO model. Additionally, the q_e,cal_ of As(V) and TC adsorption obtained from the PSO model was much closer to the experimental adsorption capacity than that of the PFO model. The above results suggest that PSO is more accurate for interpreting the adsorption behavior of CMS towards As(V) and TC. Thus, chemisorption is a limiting factor of the sorption rate during the removal of As(V) and TC [36].

To investigate the possible transportation mechanisms involved in the adsorption of As(V) and TC, the Weber–Morris intraparticle diffusion model and the Boyd model were employed to fit the adsorption data during the kinetics study (detailed information about the intraparticle diffusion kinetic model and the Boyd model is listed in Appendix A, and the results are summarized in Table 4). As shown in Figure 9A1,A2, the adsorption process of As(V) and TC onto CMS could be divided into two stages [25,39]. The first stage is commonly recognized as a mass transfer process in which As(V) or TC contaminants in solution diffuse to the external surface of CMS. Contaminants of As(V) and TC are adsorbed on the external surface of CMS through adsorptive sites. The intraparticle diffusion constant k_id_ represents the diffusion rate, which is affected by the contaminant species, concentration, and so on. It was noticed that the diffusion rate of As(V) was higher than that of TC based on a higher value of k_1d_ during the first stage (Table 4). Although the electrostatic attraction between CMS and TC contributed to the diffusion rate of TC, the As(V) existed as highly mobile ions with small molecule weights. Thus, As(V) exhibited a higher diffusion rate than organic TC, which possessed a high molecule weight during the first stage. Comparatively, the second stage was ascribed to the diffusion of As(V) and TC from the exterior surface to the internal surface and pores, which was controlled by intraparticle diffusion. In this stage, the k_2d_ of As(V) diffusion was also higher than that of TC, indicating that As(V) possessed a higher diffusion rate than TC. The molecule weight of TC is much higher; thus, it was more difficult for TC to diffuse through boundaries, resulting in a low diffusion rate of TC. Based on the above analysis, the adsorption of CMS towards As(V) has a higher rate in both external mass transfer and intraparticle diffusion, resulting in a faster adsorption towards As(V).

As the adsorption process consists of external mass transfer and intraparticle diffusion, it was necessary to judge which one exerted a greater influence on the adsorption rate of As(V) and TC. Subsequently, the Boyd model was adopted to further analyze the adsorption kinetic data [40]. As shown in Figure 9B1,B2, the fitted plot of As(V) adsorption was nearly a straight line according to a high coefficient (R^2^ = 0.95), suggesting that the external mass transfer process delivered a greater influence of As(V) adsorption onto CMS, whereas the fitted plot of TC adsorption was not a straight line. The plot could be separated into two straight lines with coefficients of 0.98 and 0.98, respectively. In this situation, the fitted plot of TC adsorption revealed that external mass transfer exhibited a weak influence on the TC adsorption rate, and it was intraparticle diffusion that mainly dominated the sorption rate of TC onto CMS.

### 3.7. Adsorption Mechanism of CMS towards As(V) and TC

To identify the carbonaceous species and chemical environment, XPS spectra of the fabricated pure hydrothermal carbon (Table 1, C1) are illustrated in Figure 10. The XPS survey spectrum (Figure 10A) showed that significant N1s, O1s, and C1s characteristic peaks appeared, which demonstrated the successful hydrothermal reaction of ammonia with glucose [41]. Additionally, the atomic ratios of C, N, and O elements were 76.47%, 14.23%, and 9.30%, respectively, suggesting that the fabricated hydrothermal carbon had abundant N-containing functional groups. As illustrated in Figure 10B, the C element mainly existed as C-C/C=C/CH_x_, C=N/C-OR, C-N/C-OH, and COOR at the binding energy values of 284.6 eV, 285.7 eV, 287.1 eV, and 289 eV, respectively [42]. The binding energies and content of C and N species are summarized in Table 5. N 1s spectrum (Figure 10C) analysis showed that the N 1s spectrum could be fitted into three peaks at binding energies of 399 eV, 400.3 eV, and 401.4 eV, which corresponded to pyridinic-N, pyrrolic-N, and graphitic-N [42], respectively, while the O1s XPS spectrum (Figure 10D) showed that the O element existed in the form of C=O bonds and C-O bonds. The existence of pyridinic-N, pyrrolic-N, and graphitic-N revealed that the N element can be doped into carbon species through hydrothermal reactions, and hydrothermal carbon with abundant C- and N-containing groups is favorable for the removal of contaminants in sewage. Thus, the combination of hydrothermal carbon with magnesium silicate can provide C- and N-containing groups for CMS.

In order to determine the role of C- and N-containing groups in the adsorption of CMS towards As(V) and TC, XPS measurements were conducted, and the results are displayed in Figure 11. In addition to the XPS signals of Mg, Si, and O elements, C1s and N elements were also observed in the XPS survey of CMS before the adsorption (Figure 10A,B). Considering the peak intensity of C and N elements in the XPS survey, the C and N contents in CMS were much lower than those of the fabricated N-doped hydrothermal carbon. After the adsorption of As(V), significant peaks of As 2p and As 3d were observed in the XPS survey of CMS, revealing that As(V) was successfully adsorbed onto CMS. As for TC adsorption, the intensity of the Mg XPS signal significantly decreased in the XPS survey of CMS, indicating that the adsorption of TC consumed Mg-containing adsorption sites on the surface of CMS.

Compared with pure hydrothermal carbon (Table 1, C1), the C and N species and binding energy in CMS remained unchanged, whereas the contents of C and N species exhibited a noticeable change, which may be attributed to the co-existence of magnesium silicate in the fabrication process. The binding energy and content of C and N species in CMS are summarized in Table 5. The content of pyridinic-N, pyrrolic-N, and graphitic-N changed after the adsorption of CMS towards As(V) and TC, suggesting that N species participate in the adsorption of As(V) and TC [42]. As for the adsorption of As(V) (Figure 11A2), the bigger change in the content of pyridinic-N revealed that pyridinic-N in N species mainly contributed to the adsorption of As(V). As for the adsorption of TC (Figure 11B2), the contents of pyridinic-N and graphitic-N decreased, indicating that pyridinic-N and graphitic-N participated in the adsorption of TC. The TC molecule is a π-electron-rich donor due to its structure of a benzene ring and C- and N-containing groups. Graphitic-N is considered to be an efficient π-electron acceptor and can adsorb TC molecules through π–π electron donor–acceptor (EDA) interactions [5,22,42]. The pyridinic-N connects TC molecules through hydrogen bonds, realizing the adsorption of TC [5,22,42]. As illustrated in Figure 11A3, the contents of COOR and C-OH/C-N groups significantly decreased after As(V) adsorption. The functional groups of COOR and C-OH/C-N connected with As(V) through complexation, resulting in a decrease in the contents of COOR and C-OH/C-N groups [22,36]. In the adsorption of TC, the contents of C-OR/C=N and C-OH/C-N groups decreased. The C=N and C-N bonds corresponded to graphitic-N and pyridinic-N, which enabled the TC removal with π–π EDA interactions and hydrogen bond effects [42]. In summary, XPS analysis confirms the important role of C and N species in the adsorption of CMS towards As(V) and TC. With C- and N-containing groups, CMS can effectively adsorb the contaminants of As(V) and TC from sewage.

The FT-IR technique was employed to analyze the role of functional groups and adsorptive sites in the adsorption of CMS towards As(V) and TC. According to XRD and NMR analyses, CMS was composed of Mg-chlorite and N-doped hydrothermal carbon, suggesting CMS possesses Mg-adsorptive sites and C- and N-containing groups. As illustrated in Figure 12a, the adsorption peaks at 1016, 3439, and 3679 cm^−1^ were attributed to the Si-O-Mg bond, a dissociative hydroxyl group, and Mg-OH, respectively. The Si-O-Mg bond and Mg-OH were attributed to magnesium silicate in CMS, whereas the dissociative hydroxyl group consisted of a hydroxyl group from magnesium silicate and C-OH from carbonaceous species. The component of magnesium silicate possessed limited variety of functional groups, while the component of N-doped hydrothermal carbon had various active functional groups. The peaks at 1080 and 1240 cm^−1^ were attributed to the vibration of C-O and C-N bonds, respectively. The above peaks in the FT-IR spectrum were affected by the large and wide peak of the Si-O-Mg bond. The peaks at 1450, 1625, and 1653 cm^−1^ corresponded to the COOR group, C=N bond, and C=O/C=C bond, respectively. The latter two peaks exhibited as one peak in the FT-IR spectrum due to the relatively close wavelengths.

After the adsorption of As(V), the peak intensities of the Si-O-Mg bond, COOR, C=N, C=O, dislocated hydroxyl group, and Mg-OH significantly decreased, indicating that the above adsorptive sites participated in the adsorption of CMS towards As(V), which is consistent with the XPS analysis results. This confirmed the contribution of C- and N-containing groups. The intensity decrease in the Si-O-Mg bond and Mg-OH was attributed to the formation of amorphous magnesium arsenate complex ((Mg-O)AsO_3_^−^) [22,43]. Moreover, the amorphous metal complex and As(V) could also be adsorbed by COOR, C=N, and C=O bonds, resulting in a decrease in the peak intensity through hydrogen bond interactions. Based on the above analysis, multi-adsorptive sites participate in the adsorption of CMS towards As(V).

Figure 12c illustrates the FT-IR spectrum of CMS after TC adsorption. It was observed that the peaks of Mg-OH and N-H bond disappeared while the peak intensities of the C-N and Si-O-Mg bonds decreased, which was attributed to the complexation of Mg-containing adsorptive sites with TC molecules into magnesium tetracycline complex (MgTC) [30]. Since TC is a π-electron-rich donor and possesses hydroxyl, carbonyl, and amino groups, it can donate numerous electrons to Mg-containing groups to form a MgTC complex via cation–π bridge and cation–n bridge interactions [22,30]. Then, N-H bonds in graphitic-N and pyridinic-N connect with MgTC or TC through π–π EDA interaction and hydrogen bonds. As the TC molecule possessed abundant COOR, C=O, C=C, and C-O bonds, the observable increase in peak intensity of the above C-containing groups provided solid evidence of the high adsorption capacity of CMS towards TC. Figure 12d presents the FT-IR spectrum of the synthesized MgTC. The adsorption peaks at 1475 and 846 cm^−1^ were observed in MgTC and CMS after the adsorption, suggesting that the MgTC complex was adsorbed onto CMS. According to XRD and FT-IR analysis, CMS is composed of N-doped hydrothermal carbon and Mg-rich magnesium silicate, which provide CMS with abundant Mg-adsorptive sites and C- and N-containing groups. Due to the abundant adsorptive sites and functional groups, CMS possesses a satisfying adsorption capacity towards As(V) and TC.

### 3.8. Synergism in Construction and Adsorption

In the construction of CMS, hydrothermal carbon and magnesium silicate play different roles. They synthetically endow CMS with abundant C- and N-containing groups and high surface area. Fabricated 3D magnesium silicate possesses a high specific surface area [21], which mainly serves as a frame in the construction of CMS. As discussed in the FT-IR analysis, hydrothermal carbon provides abundant C- and N-containing groups, which make the composite CMS more adsorptive [41]. Especially, the doping of the N element leads to more defects and higher surface energy of hydrothermal carbon, which is favorable to adsorption reactions [36,42]. The dopant of the N element exists as pyridinic-N, pyrrolic-N, and graphitic-N, which could connect the TC molecule with π–π EDA interactions. Through combination, CMS inherits a 3D structure with a large surface area from magnesium silicate and abundant C- and N-containing groups from N-doped hydrothermal carbon. Thus, they synergistically participate in the construction of 3D CMS with a high surface area and an abundance of various functional groups.

For a better understanding of synergism, the construction mechanism of CMS and its synergistic adsorption behaviors towards As(V) and TC are schematically depicted in Figure 13. In the adsorption of CMS towards As(V) and TC, various functional groups also exhibit synergistic effects in the adsorption processes. In the adsorption of As(V), Mg-containing groups and C- and N-containing groups synergistically participate in the removal of As(V) through different mechanisms. The Si-O-Mg bonds as well as the Mg-OH group react with As(V) to form an amorphous magnesium arsenate complex. C- and N-containing groups connect with the amorphous complex through adsorption and chelation. In the adsorption of TC, Mg-containing groups chelate with TC to form MgTC, while C- and N-containing groups connect with TC or MgTC through π–π EDA interactions and hydrogen bonds. Thus, both Mg-containing groups and C- and N-containing groups contribute to the adsorption of TC by CMS.

## 4. Conclusions

In summary, a 3D porous N-doped CMS is fabricated using a one-step hydrothermal route with ammonia and glucose as N and C sources, respectively. Through the synergic combination of hydrothermal carbon and magnesium silicate, the obtained CMS possesses a high surface area of 201 m^2^/g and abundant C- and N-containing groups. The existence and morphology formation mechanisms of C and N species are thoroughly investigated. The maximum adsorption capacities of CMS towards As(V) and TC are 498.75 mg/g and 1228.5 mg/g, respectively. The adsorption isotherm fitting results imply that the adsorption types of As(V) and TC are both monolayer adsorption. The kinetics analysis indicates that the external mass transfer process has a greater influence on As(V) adsorption onto CMS, whereas intraparticle diffusion mainly dominates the sorption rate of TC adsorption onto CMS. Various functional groups synergically participate in the adsorption of As(V) and TC. The adsorption mechanism of CMS towards As(V) includes complexion and hydrogen bond interactions, while the adsorption of TC by CMS is mainly due to complexion caused by cation–π bridges, cation–n bridges, π–π EDA interactions, and hydrogen bonds. The facile synthesis procedure and good adsorption capacity make CMS a promising adsorbent in practical environmental remediation.

## Figures and Tables

**Figure 1 materials-16-05338-f001:**
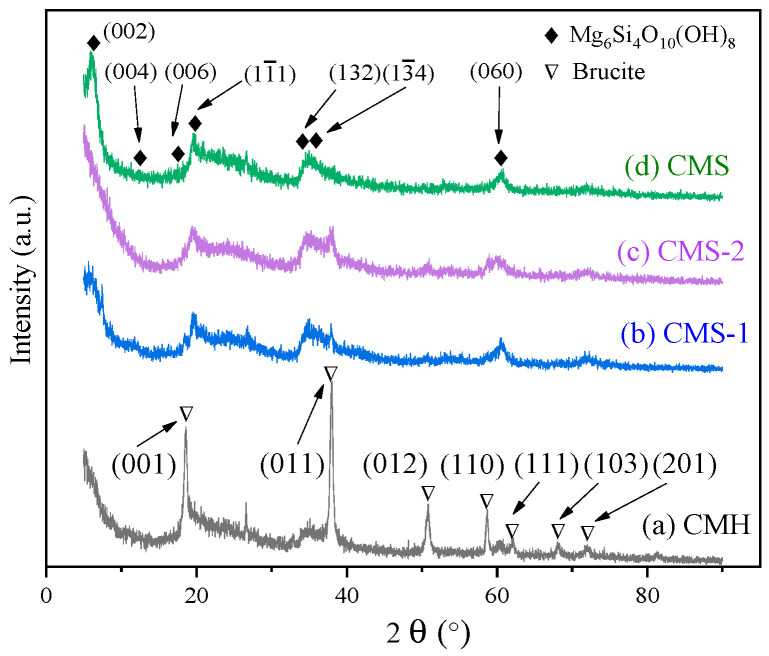
XRD patterns of the obtained carbonaceous composites with different raw material ratios.

**Figure 2 materials-16-05338-f002:**
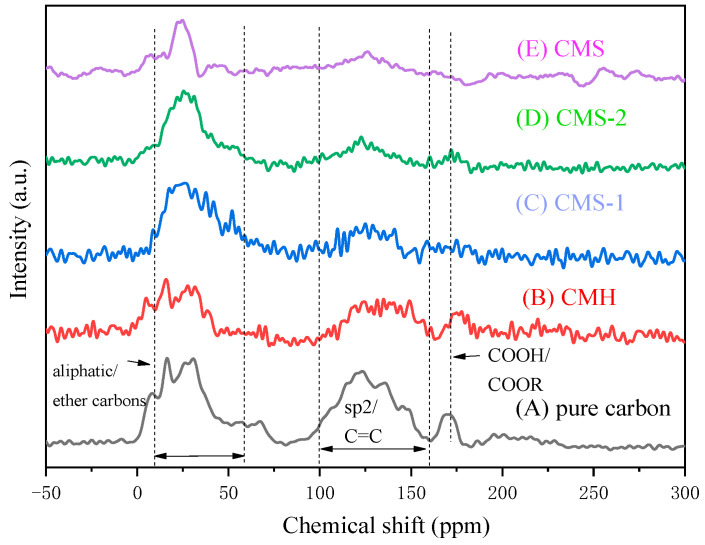
^13^C solid-state NMR of (**A**) pure carbon, (**B**) CMH, (**C**) CMS-1, (**D**) CMS-2, and (**E**) CMS.

**Figure 3 materials-16-05338-f003:**
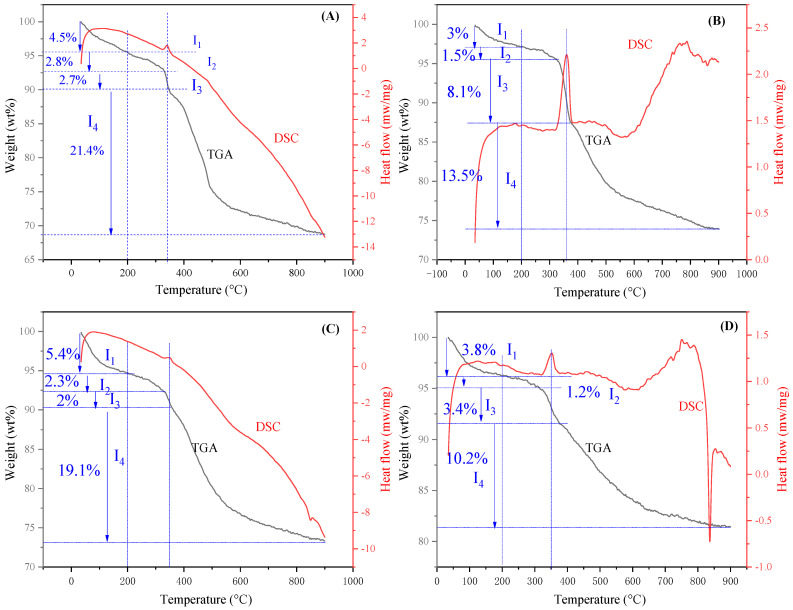
TG-DSC curves of sample CMH (**A**), CMS-1 (**B**), CMS-2 (**C**), and CMS (**D**).

**Figure 4 materials-16-05338-f004:**
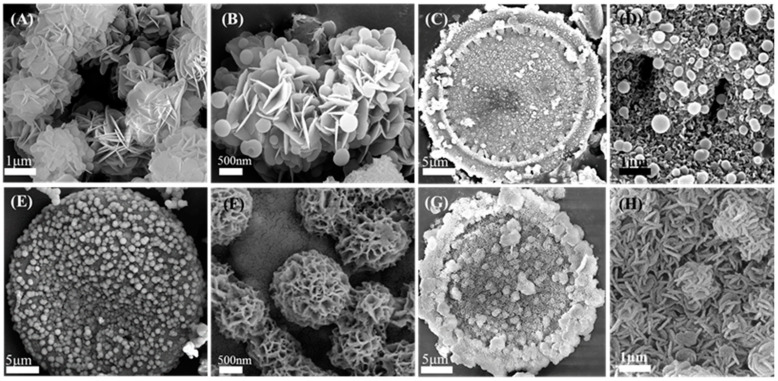
SEM images of samples obtained under different fabrication process: (**A**,**B**) CMH, (**C**,**D**) CMS-1, (**E**,**F**) CMS-2, and (**G**,**H**) CMS.

**Figure 5 materials-16-05338-f005:**
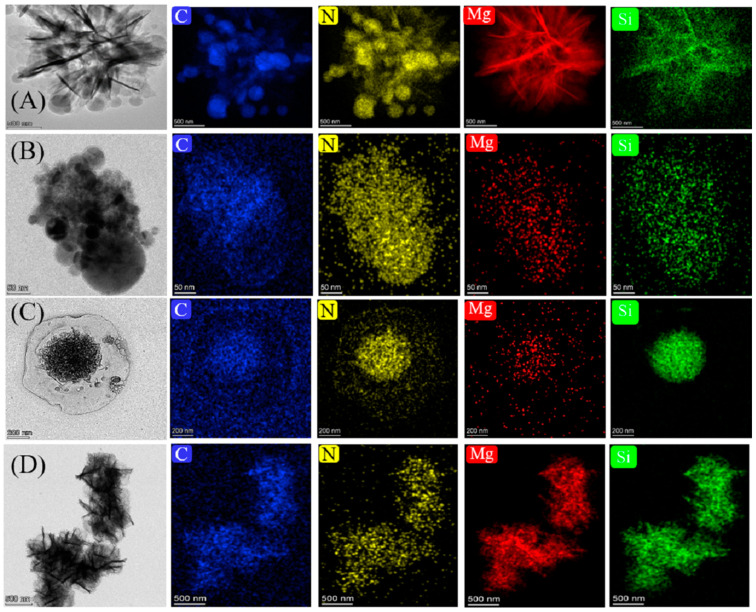
TEM images of sample (**A**) CMH, (**B**) CMS-1, (**C**) CMS-2, and (**D**) CMS and C, N, Mg, and Si element distribution, respectively.

**Figure 6 materials-16-05338-f006:**
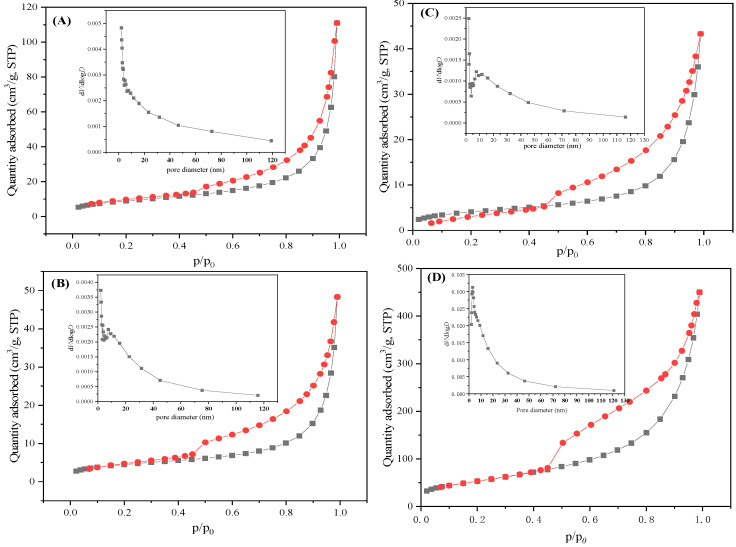
N_2_ adsorption–desorption isotherms of CMH (**A**), CMS-1 (**B**), CMS-2 (**C**), and CMS (**D**) (the red line is adsorption branch and the black line is desorption branch, the inner is pore-size distribution).

**Figure 7 materials-16-05338-f007:**
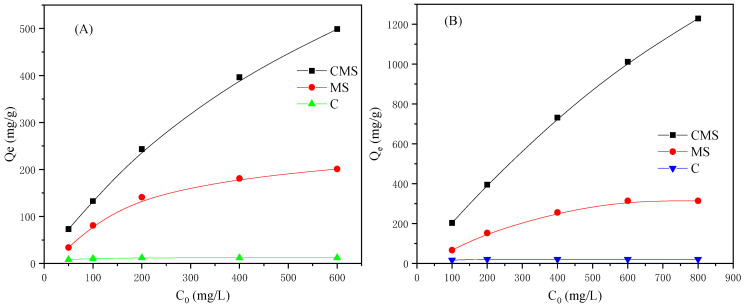
Adsorption capacity of CMS, MS, and pure carbon towards As(V) (**A**) and TC (**B**).

**Figure 8 materials-16-05338-f008:**
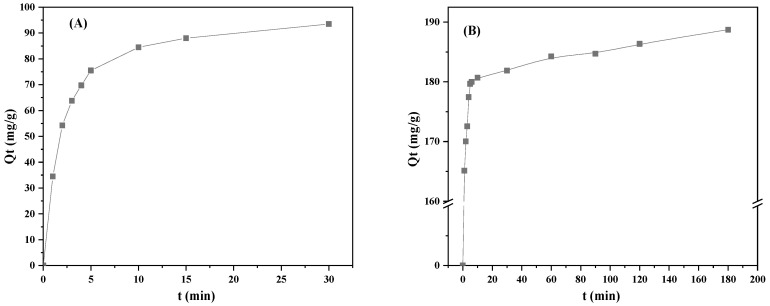
Effect of adsorption time on the adsorption capacity of CMS towards As(V) (**A**) and TC (**B**).

**Figure 9 materials-16-05338-f009:**
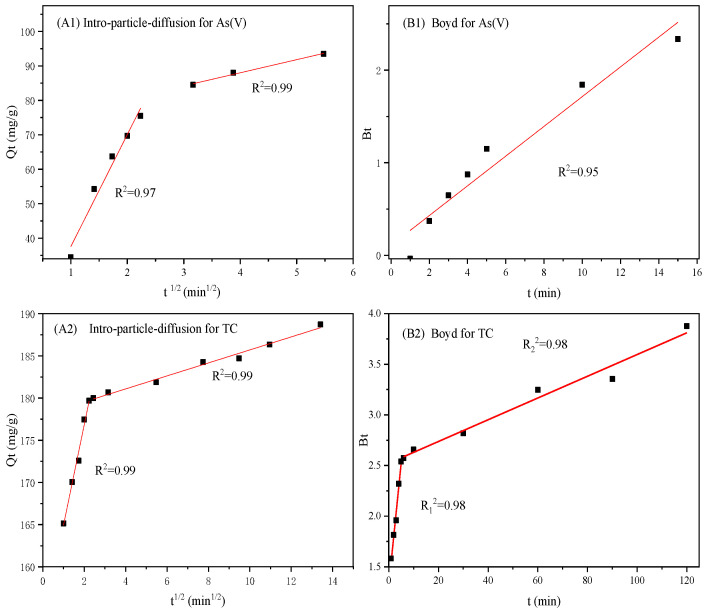
Kinetics models of (**A1**,**A2**) intraparticle diffusion and (**B1**,**B2**) Boyd for As(V) and TC, respectively.

**Figure 10 materials-16-05338-f010:**
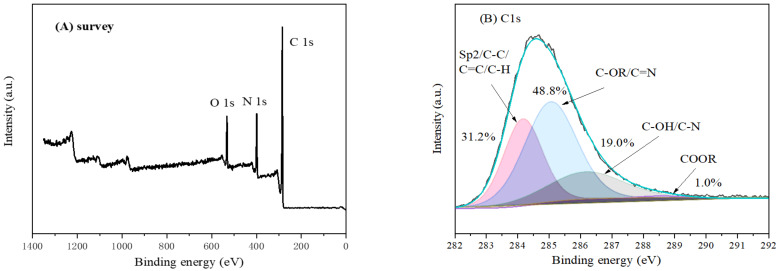
XPS spectra of pure hydrothermal carbon (**A**) survey, (**B**) C 1s, (**C**) N 1s, and (**D**) O 1s.

**Figure 11 materials-16-05338-f011:**
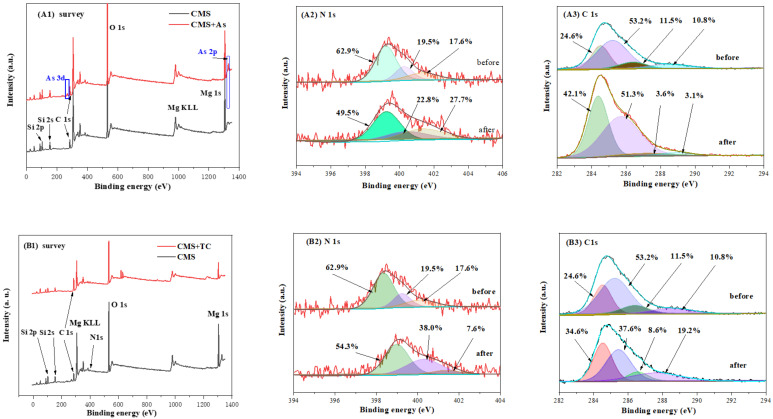
XPS spectra of CMS before and after As(V) and TC adsorption for (**A1**,**B1**) survey, (**A2**,**B2**) N1s spectra, and (**A3**,**B3**) C 1s spectra.

**Figure 12 materials-16-05338-f012:**
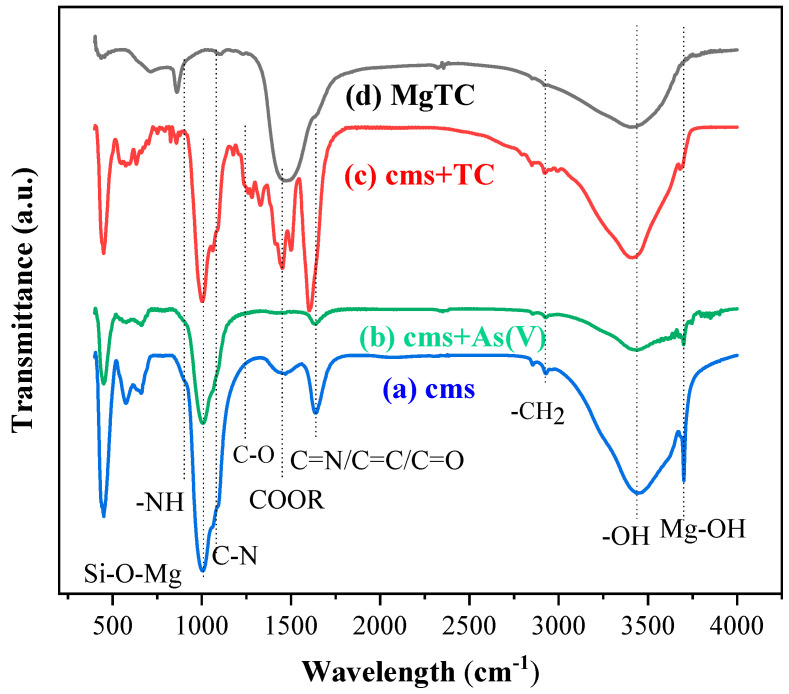
FT-IR spectrum of sample CMS (**a**) after As(V) (**b**), TC (**c**) adsorption and MgTC complex (**d**).

**Figure 13 materials-16-05338-f013:**
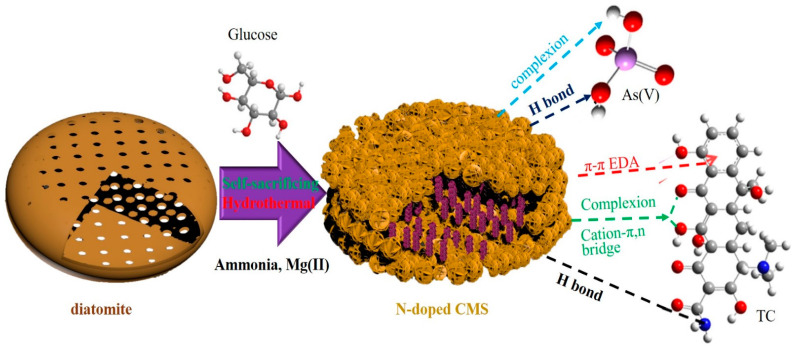
Diagrammatic illustration of CMS construction and its synergistic adsorption behavior towards As(V) and TC.

**Table 1 materials-16-05338-t001:** Raw materials used in the fabrication process.

Item		Glucose (g)	Ammonia (mL)	Ethanol (mL)	Diatomite (g)	MgCl_2_·6H_2_O (g)	S_BET_ (m^2^/g)	Sample
No.								
1#		1	20	0	1	5	32	CMH
2#		1	40	0	1	5	27.7	CMS-1
3#		1	40	5	1	5	33.7	CMS-2
4#		0.5	40	5	1	5	201	CMS
5#		1	20	0	0	0	/	C2
6#		1	20	5	0	0	/	C1
7#		0	40	0	1	5	337	Magnesium silicate [21]

**Table 2 materials-16-05338-t002:** Comparison of the As(V) and TC adsorption capacity of CMS with various adsorbents reported in the literature.

Adsorbent	pH	Q_max(As(V))_ (mg/g)	Q_max(TC)_ (mg/g)	Ref.
Magnetic graphene oxide	5	-	149	[31]
Yttrium-doped iron oxide	7	170.4	-	[38]
N-doped carbon	8	-	339	[32]
Y-GO-SA	5	273	478	[22]
Mg–N-co-doped lignin	7	687	-	[36]
Fe-doped activated carbon	4.35	-	625	[33]
Boron nitride with N-defects	7	-	1101	[34]
CMS	7	498.75	1228.5	Present work
Fe/porous carbon	7	-	1301	[35]
Covalent organic frameworks	7	787	-	[37]

**Table 3 materials-16-05338-t003:** Fitting results of pseudo-first-order and pseudo-second-order models.

Model		Pseudo-First-Order				Pseudo-Second-Order			
Species		q_e,exp_ (mg/g)	q_e,cal_ (mg/g)	k_1_ (min^−1^)	R^2^	q_e,exp_ (mg/g)	q_e,cal_ (mg/g)	k_2_ (min^−1^)	R^2^
As(V)		93.5	50.995	0.16055	0.95	93.5	98.619	6.023·10^−3^	1.00
TC		188.725	13.17	0.0149	0.79	188.725	187.970	0.0108	1.00

**Table 4 materials-16-05338-t004:** Fitting results of Weber–Morris model and the Boyd model.

Model		Weber–Morris						Boyd
Species		K_1d_ (mg·g^−1^·min^−0.5^)	R_1_^2^	C_1_	K_2d_ (mg·g^−1^·min^−0.5^)	C_2_	R_2_^2^	R^2^
As(V)		32.508	0.97	5.052	3.810	72.777	0.99	0.95
TC		11.875	0.99	153.059	0.7702	178.004	0.99	0.98/0.98

**Table 5 materials-16-05338-t005:** Concentration of C and N in different samples.

Bond/Species	Binding Energy (eV)	Concentration (%)			
		Pure C	CMS	CMS after As(V) Adsorption	CMS after TC Adsorption
C-C/C=C/CHx	284.6	31.2	24.6	42.1	34.6
C-OR/C=N	285.7	48.8	53.2	51.3	37.6
C-OH/C-N	287.1	19.0	11.5	3.6	8.6
COOR	289	1.0	10.8	3.1	19.2
Pyridinic-N	399	65.0	62.9	49.5	54.3
Pyrrolic-N	400.3	27.3	19.5	22.8	38.0
Graphitic-N	401.4	7.8	17.6	27.7	7.6

## Data Availability

Not applicable.

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
