# Peer review of "Fabrication of Nitrogen-Doped Carbon@Magnesium Silicate Composite by One-Step Hydrothermal Method and Its High-Efficiency Adsorption of As(V) and Tetracycline"

_materials, 2023, doi:10.3390/ma16155338_

Round 1

Reviewer 1 Report

Manuscript ID: materials-2482621

Title: Fabrication of nitrogen-doped carbon@magnesium silicate composite by one-step hydrothermal method and its high-efficiency adsorption of As(V) and Tetracycline

Journal: Materials

Recommendation: Major revision

Comments:

-          Most of the figures’ quality not up to the mark. The authors are suggested to provide the high-quality figures for fallowing figure numbers, such as Figure 2, Figure 3 and from Figure 6 to Figure 13.

-          In page 14, the figure captions are missing.

-          The authors are recommended to control the similarity index of entire manuscript. The major similarities matching with the already published contends in “A diatomite-template self-sacrificing route for Mg-chlorite and its adsorption behavior towards Pb(II)/Cd(II)” (https://doi.org/10.1016/j.surfin.2022.101775)

-          Grammatical errors and punctuation mistakes need to be thoroughly addressed in the manuscript

-          Reusability and Reproducibility are importance analysis for adsorption study. So the authors are recommended to carry out the ‘Reusability’ and ‘Reproducibility’ studies.

Grammatical errors and punctuation mistakes need to be thoroughly addressed in the manuscript

Reviewer 2 Report

Comments to the authors

In this manuscript, the authors reported the fabrication of N-Doped Carbon@Magnesium Silicate Composite via hydrothermal approach and evaluated its potential for the adsorption of As(V) and Tetracycline. Fabricating an efficient adsorbent for the effective decontamination of organic and inorganic pollutants has received increased attention in the last decades. In this work, an approach was introduced in order to attain his goal. The manuscript language is good and the research work is supported with scientific references. However, I suggest minor revision for quality enhancement, which are given bellow.

1.      The manuscript contains a large number of grammatical and typing mistakes which should be carefully read and the language need improvement.

2.      The authors should properly correlate the XRD and FTIR data with the results presented in the manuscript.

3.      The quality of figure 2, 3, 6, 8, 10 and 11 are very poor. Redrawn all low-quality figures with improved quality.

4.      Figure 4G and H were not discussed in the manuscript text. Why?

5.      There is no need to mention (A) in figure 7.

6.      How the removal of both the pollutants are represented through single graph (figure 7).

7.      What does “Containments” mean in the whole manuscript> I think this is contaminants. Clarify this.

8.      Why figure 9 has no caption?

9.      Explain the novelty of your work.

10.  Mention the scientific adsorption results obtained in the abstract and conclusion.

Minor editing of English language required

Reviewer 3 Report

The manuscript describes the synthesis of nitrogen-doped carbon@magnesium silicate and its application as an adsorbent for tetracycline (TC) and arsenic. The materials studied were characterized with different methods to evaluate their composition and mechanism of action as adsorbents.

In the introduction, the authors state that carbonaceous adsorbents have attracted much attention in the removal of TCs and arsenic contained sewage, yet do not present any references. I suggest to add suitable references.

The authors indicate the drawbacks of carbonaceous adsorbents, but references provided are still only linked to selective phosphate adsorbents and Cr(VI) adsorbents (ref 19,20).

Why the fabrication of the materials was provided by hydrothermal treatment only at 180 oC for 8 h with no optimization studies provided?

In Figure 1, the description of a, b c d is missing.

In the experimental part, it should be listed what CMH, CMS-1 and CMS-2, CMS indicate.

In the results and discussion I suggest to add a description of the synthetic method for clarity.

The authors indicate that Fig. 4 shows SEM images of samples obtained under different fabrication process – what was the fabrication process in this case?

Why Fig.8a does not start with 0 (time)?

Instead of writing that the peak height of N element in XPS analysis was approximately the same as O element and slightly lower than that of C element, which suggests the fabricated hydrothermal carbon owned abundant N containing functional groups, the authors could provide %contribution of specific elements in the material.

Minor suggestions: fig2 is not very readable with e) on the top. Magnetic graphene oxide instead of magnetic grapheme oxide.

The manuscript is understandable, with minor editting required.

Round 2

Reviewer 1 Report

The authors are responded to most of the comments, however, reviewer did not find any respective supporting data for Reproducibility (#Referee 1, comment 4).   The authors are recommented to include the Reproducibility results in revised manuscript before it is accepted.

Reviewer 3 Report

The authors addressed the concerns raised, therefore I recommend it for publication. 

Minor grammar issues have been noticed.
